# Visual Impairment and Blindness among Patients at Nigeria Army Eye Centre, Bonny Cantonment Lagos, Nigeria

**DOI:** 10.3390/healthcare10112312

**Published:** 2022-11-18

**Authors:** Ngozika Esther Ezinne, Oluwaseun Shittu, Kingsley Kene Ekemiri, Michael Agyemang Kwarteng, Selassie Tagoh, Grace Ogbonna, Khathutshelo Percy Mashige

**Affiliations:** 1Optometry Unit, Department of Clinical Surgical Sciences, Faculty of Medical Sciences, University of the West Indies, St. Augustine 685509, Trinidad and Tobago; 2Department of Optometry, Faculty of Health Science, Madonna University, Elele 510242, Rivers State, Nigeria; 3Discipline of Optometry, School of Health Sciences, University of KwaZulu-Natal, Durban 4041, South Africa; 4Department of Optometry, Faculty of Science and Engineering, Bindura University of Science Education, Bindura Private Bag 1020, Zimbabwe; 5School of Optometry and Vision Science, University of Auckland, Auckland 1010, New Zealand; 6Department of Optometry, Faculty of Health Science, Mzuzu University, Mzuzu P.O. Box 201, Malawi

**Keywords:** visual impairment, blindness, Lagos State, Nigeria, cataract, glaucoma, refractive error

## Abstract

**Background:** Visual impairment (VI) is a public health problem that can affect an individual’s social wellbeing. The study aims to determine the distribution and causes of vision impairment (VI) and blindness among patients at Nigerian Army Eye Centre Lagos, Nigeria. **Method:** An institutional cross-sectional study was conducted, and a systematic random sampling technique was used to enrol study participants from their medical records. Information about their demography, presenting visual acuity (VA), best corrected visual acuity and cause of VI and blindness, were retrieved. **Result:** A total of five hundred (500) medical records of patients aged from 4 to 96 years, with a mean age of 54.07 ± 21.43 years, were considered for the study. Among the participants, more than half were males (51.2%) and ≥60 years (53.0%). A large (47.2%) proportion of the patients had moderate VI at the time of presentation, followed by blindness (22.0%). The major cause of blindness was cataract, while glaucoma and refractive error were the major causes of VI. Blindness and VI were significantly associated with the type of VI before and after the provision of intervention (*p* < 0.05) across different age groups (children, youths, adults, elderly) with an adjusted *p* < 0.003 after an intervention. **Conclusions:** Cataracts, glaucoma and uncorrected refractive error (URE) were the major causes of VI and blindness in Lagos State. VI was more prevalent in males than females; however, there was no significant difference between the two proportions. The prevalence of VI among age groups was more significant for those 60 years and above. Early screening for the detection and management of cataract, URE and glaucoma is highly advised to reduce the burden of VI.

## 1. Introduction

Visual impairment (VI) is a significant public health issue around the world because it has a detrimental impact on a person’s psychosocial and economic well-being, as well as the well-being of their family, community, and the country as a whole [1,2]. In children, this leads to lifelong consequences, including a lower level of educational achievement [3]. Previous studies [4,5,6,7,8] showed that about 36 million individuals are blind, 217 million have a moderate-to-severe visual impairment, and 253 million are visually impaired. A recent World Health Organization (WHO) report showed that the population size of the visually impaired has increased to 2.2 billion, and 1 billion cases of VI could have been prevented [5,9]. Visually impasired people make up more than 90% of the population in developing nations and are 50 years and above [8]. Age, religion, and ethnicity were associated with VI [4,10].

There are 88.4 million cases of refractive error, 94 million cases of cataract, 7.7 million cases of glaucoma, 4.2 million cases of corneal opacities, 3.9 million cases of diabetic retinopathy, 2 million cases of trachoma, and 3.8 million cases of near-VI brought on by untreated presbyopia (826 million). These were reported as the major causes of VI and blindness globally [5,11] In Nigeria, cataract was recorded as the major cause of VI, constituting 45.3% and 43.0% of VI and blindness, respectively [2,11].

According to estimates, low- and middle-income nations have a four times higher prevalence of distance VI than high-income countries [3]. Approximately 4.25 million adults aged ≥40 were visually impaired in Nigeria [12]. With a population of 207 million, Nigeria is the most populated country in Africa. [13]. The prevalence of blindness and VI in Nigeria were reported to be 6.1% and 4.2%, respectively [12], which was significantly higher than reported globally [5]. Studies [10,12,14,15] conducted in different parts of Nigeria indicated that the prevalence of blindness and VI is highest in the north–east (8.2% and 6.9%) and lowest in the south–west (2.8% and 3.3%) part of Nigeria.

Although the national prevalence of VI and blindness survey recorded the lowest prevalence of VI and blindness in Southwestern Nigeria, there is still a paucity of literature on studies reporting VI and blindness estimates in the southwestern part of Nigeria. No study has been done to determine the distribution and causes of VI and blindness in Lagos, even though Lagos is the most populous city in Africa, with a population of over 21 million [16]. This study aims to ascertain the distribution and causes of VI and blindness among patients that visited the Nigeria Army Eye Centre, Lagos. The findings from this study would aid in the design of strategic plans for eliminating and reducing the VI burden in Lagos State.

## 2. Method

### 2.1. Study Design and Setting

A one-year institution-based cross-sectional study was conducted to determine the distribution and causes of VI and blindness at the Nigeria Army Eye Centre, Lagos. The Eye Centre is located in the Bonny camp a military cantonment in Ikoyi, Victoria Island, Lagos state with a few civilian and majority of the residents in the military. The centre is a sub-unit of the Nigeria Military Hospital Lagos State. It has clinics for paediatrics, anterior segment, glaucoma, and optometry, among other subspecialties, to provide the residents of the cantonment and its surroundings with tertiary eye-care services.

### 2.2. Study Population and Sampling Technique

The study population were patients attending the Nigeria Army Eye Centre in Lagos State, Nigeria, between January 2016 and December 2016—an estimated total population of about 5500 patients. This population was used to determine the sample size using a single-population proportion formula. VI was considered as a 50% proportion of the population; the sample size was calculated using 95% confidence interval, 3% margin error, and 10% non-response rates. Hence, the sample size was 664. Patients of all age groups that visited this eye centre within the study period presenting a visual acuity of less than 6/12 at distance in the better eye were eligible for the study. The eye centre’s patient registration logbook was used to find study participants presenting less than 6/12 visual acuity at distance in the better eye; the first patient was the 5th medical record number entered; registration logbooks with incomplete data were excluded.

### 2.3. Inclusion and Exclusion Criteria

Medical records of all individuals that visited the eye centre within the study period were included. Individuals whose medical records had incomplete information, including presenting visual acuity (VA) and cause of reduced VA, were excluded from the study.

### 2.4. Data Collection Procedure

Electronic records, indexing, and database registries are absent from the eye centre. All medical records of people who visited the eye centre during the study period were first requested from the appropriate authority and then recovered from the archives with the help of the eye centre secretary, because the eye centre still uses a hard copy system to preserve patient records. Age, gender, clinical variables such as Snellen VA (presented and best corrected), and a list of every ocular diagnosis recorded in the patient’s medical file were recorded. The presenting and post-treatment VA on the better eye were recorded for each individual. An HOTV chart was used to determine the VA for children under five. Only distance VA was considered for this research. Data were recorded on a data extraction sheet, and the cause of VI and blindness was determined from the individual’s medical records. For patients with co-morbidities, the condition that significantly affects vision was used as the cause of VI/blindness.

### 2.5. Ethical Considerations

The study was conducted per Helsinki’s declaration and approved by the institutional review board of Madonna University, Elele Campus, Nigeria (MAU/SREC/A/055/07/2018). Permission to access patients’ medical records was obtained from the relevant authorities. The privacy and confidentially of all participants’ information were maintained.

### 2.6. Data Analysis

Statistical Package for Social Sciences (SPSS) version 24 was used to analyse the data (IBM Corp., Armonk, NY, USA, 2012). Descriptive statistics such as frequency distribution and central tendency measurements were used to summarise the descriptive portion of the investigation. To identify the variables connected to VI and blindness, the Pearson Chi-squared test was utilised. Statistical significance was defined as a *p*-value of 0.05.

### 2.7. Operational Definition

Participants in the study had their vision status determined using the WHO categories of VI. “Blindness was defined as a presenting VA of less than 3/60 in the better eye. VI was defined as presenting VA of at least 3/60 but less than 6/12 in the better eye” [7].

## 3. Result

### Demographic Profiles of the Participants

Five hundred and fifty medical records were selected, but 50 were excluded due to incomplete records. A total of five hundred (500) medical records of patients ranging in age from 4 to 96 years, with a mean age of 54.07 ± 21.43 years, were considered for the study. The majority (53.0%) of the participants were aged 60 and above. There were more males (51.2%) than females (Table 1).

## 4. Distribution of Visual Impairment According to Sex

Overall, presenting VI was more prevalent in males than females; however, there was no significant difference between the two proportions. A large (47.2%) proportion of the patients had moderate VI at the time of presentation, followed by blindness (22.0%) and severe VI (20.0%). This proportion indicates that the majority of participants reported late for intervention (Table 2). After the medical and optical interventions, the leading type of VI was mild (52.2%), followed by moderate (23.6%) and blindness (16.0%) (Table 2).

### 4.1. Distribution of VI According to Age Groups

Among the age groups, the most prevalent category of unaided VI was moderate (47.2%). The second leading category of VI among the elderly was blindness (Table 3). After the intervention, the most prevalent category was mild VI among the participants. There was an association between age group and type of VI before and after providing optical aids using Pearson Chi-square analysis (Table 3). A posthoc analysis was performed to determine the specific significant variables or sub-groups that had significant associations before optical aid; elderly and blindness had a significant association, as well as children, youth and adults with moderate VI with adjusted *p* < 0.003. The post-hoc analysis was further used to determine the significant variables or sub-groups significantly associated with optical aid. Children, youth, elderly and mild VI had a statistically significant association with adjusted *p* < 0.003.

### 4.2. Causes of VI and Blindness

Table 4 describes the causes of VI and blindness. Cataract was the primary cause of VI and blindness, which could be due to the high proportion of elderly participants in this study. The second leading cause of VI was URE, followed by glaucoma, whilst glaucoma was the second most common cause of blindness.

### 4.3. Association between Age Groups and Causes of VI and Blindness

A Pearson chi-square test was employed to evaluate the associations between the proportions of participants in age groups and the causes of VI and blindness among them. A statistically significant distinction existed between the proportions of elderly participants with cataracts compared to the other age groups (*p* < 0.05). Additionally, the number of elderly patients with refractive errors significantly differed from the other age groups. This suggests that the elderly are more likely to get cataract than URE compared to children and other age groups (Table 5).

### 4.4. Association between Sex and Causes of VI and Blindness

A Pearson chi-square test was employed to evaluate the associations between the proportions of participants based on sex and the causes of VI and blindness. No statistically significant distinction existed between the sex and the causes of VI and blindness observed in this study (*p* > 0.05), as shown in Table 6.

## 5. Discussion

Visual impairment is a public health problem that could affect an individual’s quality of life and socio-economic status. To the best of our knowledge, our research was the first to assess the distribution and causes of VI in Lagos State, Nigeria. Moderate VI was the most prevalent category of VI and more prevalent among males 60 years and above. Cataract, URE and glaucoma were the major causes of VI and blindness.

Consistent with our study findings was a record of predilection of VI among males compared to females reported in a study in Ghana. One of the reasons that a higher VI may be recorded among males compared to females is because males are more likely to be exposed to danger or engage in sight-threatening activities than females. It could also be just a reflection of the general population of Lagos state, which has more males than females [17]. Moreover, we used pre-existing participant records; it is possible that more males attended the eye clinic for treatment than females. Contrary to our findings, studies in Nigeria [18] and South Africa [19] recorded a higher prevalence among females than males. The variation in the findings could result from differences in the gender distribution in the sample population used in each study or a disproportionate sampling and inclusion of more males or females as study participants. For example, there was a slight predilection for males than females in our study, while in the study in South Africa [19], over 80% of the sampled population were females.

Globally, age has been recorded as a major risk factor for VI as the prevalence of blindness was reported to increase with age [4,20,21] and over 4.5 million Nigerians above the age of 40 were reported to be either blind or have severe VI [12]. Consistent with these studies, the distribution of different categories of VI was found to increase with age, particularly among the older population in our study, with the proportion of participants who were blind among the adult age group being almost double the proportion who were blind in all other age groups combined. Similar findings were recorded in other studies [9,19,21,22,23,24,25]. Although these studies were conducted in different settings compared to ours, the age range for adults in these studies and ours overlap; hence, a comparison of our study to these studies in terms of tested age groups is a valid one.

Most causes of VI in the current study are preventable, as evidenced by cataract and refractive error being recorded as the major causes of VI. Similar findings were recorded in Nigeria [10] and Zimbabwe [26]. The need for continuous eye health education and the the promotion of the importance of regular screenings to prevent avoidable blindness in Lagos State is highly indicated. Most of our study participants presented late to the clinics when their VI had progressed to moderate and severe VI, although their vision improved after correction despite the late presentation. This shows that interventions such as eye-screening programs and correcting refractive error for individuals in affected communities are very effective strategies for reducing VI and must be implemented in communities in which this need is present.

The high proportion of blindness recorded in our study could be due to late presentation to the clinic. Poor access to eye health-care services and lack of knowledge about the importance of regular eye screenings could be a barrier to seeking help earlier. Education on the necessity of regular eye screening for early detection of eye conditions is an important intervention that cannot be over-emphasised in this community. Additionally, ensuring that eye-care services are available, accessible and affordable is recommended to promote the use of eye-care services.

Cataract and URE were reported in similar studies in developed countries [27,28] as the major cause of VI. Similar findings were reported in the current study. This finding was not surprising, as more than half of our study participants were aged 60 years or more, and these conditions are known to be common among the aged. There is a great need for policies and programs to increase the accessibility, availability and affordability of eye-care services, including cataract surgery and the provision of spectacles. The prevalence of ARMD and other posterior segment diseases among our study participants is quite significant, even though it was lower than reports in developed countries [29,30]. There is a need for plans to implement effective VI and blindness control programs, including monitoring the progress of these ocular conditions.

## 6. Strength and Limitations

We acknowledge that our study has some inherent limitations. Our data were collected from a single centre, limiting our ability to generalise our findings to the entire population. We also note that our study used a relatively older dataset. However, this is a vital dataset that represents the only source of information on the prevalence and distribution of VI in Lagos state, as far as we know, and must, therefore, not be ignored. Despite the limitations, our study provides a valuable insight into the distribution of VI in Lagos State, since hospital-based study findings are known to reflect a condition’s genuine character as it exists in the community. Our research findings could serve as a primary baseline for investigating further VI distribution in subsequent population-based studies in Lagos state and, ultimately, in Nigeria. Additionally, the lower number of participants than the calculated sample size underpowers the study findings.

## 7. Conclusions

Cataract, glaucoma and URE were the major causes of VI and blindness in Bonny Cantonment Lagos State. VI was more prevalent in males than females; however, there was no significant difference between the two proportions. The prevalence of VI among age groups was more significant for those 60 years and above. Frequent screening for the early detection, treatment and management of cataract, URE and glaucoma is highly advised to reduce the burden of VI. Additionally, a population-based study on the distribution and causes of VI in Lagos State is recommended.

## Figures and Tables

**Table 1 healthcare-10-02312-t001:** Distribution of Age group according to sex.

Age Group	Sex		Total (%)
	Female	Male	
Children (4–17)	26	18	44 (8.8)
Youth (18–35)	33	37	70 (14.0)
Adult (36–59)	59	62	121 (24.2)
Elderly (60 and above)	126	139	265 (53.0)
Total	244 (48.8)	256 (51.2)	500 (100)

**Table 2 healthcare-10-02312-t002:** Distribution of unaided and aided visual impairment according to sex.

Variable		Sex		Total (%)	*p*-Value
		Female	Male		
Unaided VI	Mild (less than 6/12–6/18)	21	33	54 (10.8)	0.122
	Moderate (less than 6/18–6/60)	114	122	236 (47.2)	
	Severe (less than 6/60–3/60)	58	42	100 (20.0)	
	Blindness (less than 3/60—NPL)	51	59	110 (22.0)	
Aided VI	Mild (less than 6/12–6/18)	137	124	261 (52.2)	0.332
	Moderate (less than 6/18–6/60)	52	66	118 (23.6)	
	Severe (less than 6/60–3/60)	17	24	41 (8.2)	
	Blindness (less than 3/60—NPL)	38	42	80 (16.0)	
Total		244 (48.8)	256 (51.2)	500 (100)	

**Table 3 healthcare-10-02312-t003:** Distribution of unaided and aided VI according to age-group.

Variables		Age Group				Total (%)	*p*-Value
		Children	Youth	Adult	Elderly		
Unaided VI	Mild (less than 6/12–6/18)	4	2	13	35	54 (10.8)	*p* < 0.001
	Moderate (less than 6/18–6/60)	32	46	52	106	236 (47.2)	
	Severe (less than 6/60–3/60)	6	16	29	49	100 (20.0)	
	Blindness (less than 3/60—NPL)	2	6	27	75	110 (22.0)	*p* < 0.001
Aided VI	Mild (less than 6/12–6/18)	36	54	58	113	261 (52.2)	
	Moderate (less than 6/18–6/60)	6	10	30	72	118 (23.6)	
	Severe (less than 6/60–3/60)	0	3	12	26	41 (8.2)	
	Blindness (less than 3/60—NPL)	2	3	21	54	80 (16.0)	
Total		44	70	121	265	500 (100)	

**Table 4 healthcare-10-02312-t004:** Causes of visual impairment and blindness.

Causes	Subgroup	VI	Blindness (%)	Total (%)
Treatable	Cataract	152 (38.9)	59 (53.7)	211 (42.2)
	URE	105 (26.9)	2 (1.8)	107 (21.4)
Non-treatable	Glaucoma	53 (13.6)	28 (25.5)	81 (16.2)
	ARMD	12 (3.1)	1 (0.9)	13 (2.6))
	Optic atrophy	10 (2.7)	2 (1.8)	12 (2.4)
	Optic neuropathy	9 (2.3)	2 (1.8)	11 (2.2)
	Diabetic retinopathy	7 (1.8)	3 (2.7)	10 (2.0)
	Corneal scar	7 (1.8)	3 (2.7)	10 (2.0)
Others		35 (8.9)	10 (9.1)	45 (9.0)
Total		390 (100.0)	110 (100.0)	500 (100.0)

ARMD: Age-related macular degeneration.

**Table 5 healthcare-10-02312-t005:** Association between age groups and causes of VI and blindness.

Causes	Age Group				Total (%)
	Children	Youth	Adult	Elderly	
ARMD	0 _a_	1 _a_	3 _a_	9 _a_	13 (2.6)
Cataract	1 _a_	5 _a_	42 _b_	163 _c_	211 (42.2)
Corneal Scar	1 _a_	2 _a_	2 _a_	5 _a_	10 (2.0)
Diabetic Retinopathy	0 _a_	1 _a_	5 _a_	4 _a_	10 (2.0)
Glaucoma	1 _a_	5 _a_	19 _a, b_	56 _b_	81 (16.2)
Optic Atrophy	0 _a_	2 _a_	4 _a_	6 _a_	12 (2.4)
Optic Neuropathy	1 _a_	0 _a_	4 _a_	6 _a_	11 (2.2)
Others	7 _a_	9 _a_	19 _a_	10 _b_	45 (9.0)
URE	33 _a_	45 _a_	23 _b_	6 _c_	107 (21.4)
Total	44 (8.8)	70 (14.0)	121 (24.2)	265 (53.0)	500 (100)

Each subscript letter denotes a subset of AGE GROUP categories whose column proportions do not differ significantly from each other at the 0.05 level.

**Table 6 healthcare-10-02312-t006:** Association between sex and causes of VI and blindness.

Causes	Sex		Total (%)
	Female	Male	
ARMD	7 _a_	6 _a_	13 (2.6)
Cataract	98 _a_	113 _a_	211 (42.2)
Corneal Scar	6 _a_	4 _a_	10 (2.0)
Diabetic Retinopathy	6 _a_	4 _a_	10 (2.0)
Glaucoma	33 _a_	48 _a_	81 (16.2)
Optic Atrophy	3 _a_	9 _a_	12 (2.4)
Optic Neuropathy	6 _a_	5 _a_	11 (2.2)
Others	24 _a_	21 _a_	45 (9.0)
URE	61 _a_	46 _a_	107 (21.4)
Total	244 (48.8)	256 (51.2)	500 (100)

Each subscript letter denotes a subset of sex categories whose column proportions do not differ significantly from each other at the 0.05 level.

## Data Availability

The data set is available upon request from the corresponding authors.

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
