# Peer review of "Visual Impairment and Blindness among Patients at Nigeria Army Eye Centre, Bonny Cantonment Lagos, Nigeria"

_healthcare, 2022, doi:10.3390/healthcare10112312_

Round 1
Reviewer 1 Report
The article presents and analyzes the causes of visual impaiment in patients admitted in the hospital during one year. While there might be some epidemiological interest and there is some statistical analysis between the mist frequent causes and the age and sex of the patients, the article has limitted value from the ophthalmological point of view. Treatable causes of blindness, such as cataract are mixed in the same statistics with irrreversible causes, such as age related macular degenerescence and optic atrophy.
More value could be added, if the authors could provide data for visual impairment at admission and at discharge, and differentiate between treatable and non treatable causes of blindness and visual impairment.
The analysis could also provide conclusions regarding the particularities of the patients admitted in the evaluated center and recommendations for prevention.
Author Response
Reviewer 1 comments
Response to reviewer’s comments
We are grateful for the helpful feedback by the reviewers that helped us improve the quality of the manuscript. We carefully responded to all points and have modified the manuscript accordingly.
Reviewer’s comment
The article presents and analyzes the causes of visual impairment in patients admitted in the hospital for one year. While there might be some epidemiological interest and there is some statistical analysis between the mist frequent causes and the age and sex of the patients, the article has limited value from the ophthalmological point of view. Treatable causes of blindness, such as cataract are mixed in the same statistics with irreversible causes, such as age-related macular degeneration and optic atrophy.
Response
We appreciate the comment for acknowledging that the study has some epidemiological value and the relevance of some of the stats regarding the most frequent causes of VI and age and sex. The tables have been edited to address the concerns raised on treatable and non-treatable causes of VI/blindness. Table 4, line 161
Reviewer’s comments
More value could be added, if the authors could provide data for visual impairment at admission and at discharge and differentiate between treatable and non-treatable causes of blindness and visual impairment
Response
Table 3 has data on the visual impairment at admission and at discharge. Treatable and non-treatable causes have been included in Table 4, line 161.
Reviewer’s comments
The analysis could also provide conclusions regarding the particularities of the patients admitted in the evaluated center and recommendations for prevention”
Response
Recommendations for prevention have been addressed in the discussion and further in the conclusion: In the discussion: “Education on the necessity of regular eye screening for early detection of eye conditions is an important intervention that cannot be over-emphasized in this community. Also, ensuring that eye care services are available, accessible and affordable is recommended to promote the use of eye care services”. Line 222-226
“Needs for continuous eye health education and the promotion of the importance of regular screening to prevent avoidable blindness in Lagos State are highly indicated”. Line 212-214
In the conclusion: “Frequent screening for early detection, treatment and management of cataract, URE and glaucoma is highly advised to reduce the burden of VI. Also, a population-based study on the distribution and causes of VI in Lagos State is recommended”. Line 250-257
Reviewer 2 Report
“Visual Impairment and Blindness among Patients at Nigeria Army Eye Centre, Bonny Cantonment Lagos, Nigeria” is an interesting and must needed study and I would like to commend the authors for their work.
Lines 53 onwards: While the authors in the introduction section mentioned the importance of economics/occupation/education on eye health, the current manuscript has not mentioned or explored the impact economics/occupation/education on their sample size, analysis or discussion.
Table 4 mentions the causes of VI/blindness as: “Cataract, URE, Glaucoma, ARMD, Optic atrophy, Optic neuropathy, Diabetic retinopathy, Corneal scar, Others”. Majority of these disease/pathology have no relation to VI/blindness in children, as supported by the findings in Table 5. Please explore childhood related VI/blindness factors if the children (8.8%) are included in this report, i.e., amblyopia, strabismus, ocular trauma, etc.
Please elaborate in the methods and discussion section how participants were segregated with co-morbidities, as diseases such as cataract and diabetic retinopathy can occur simultaneously occur in the same person.
Lines 80-83: “This population was used to determine the sample size using a single population proportion formula. VI was considered as a 50% proportion of the population; the sample size was calculated using 95% confidence interval, 3% margin error, and 10% non-respond rates. Hence, the sample size was 664.”
Does this mean the study was underpowered as lines 122 stated that the study assessed 500 participants, if so, please mention in the limitation of the study.
Lines 115: “Operational definition: Participants in the study had their vision status determined using the WHO categories of VI. "Blindness was defined as a presenting VA of less than 3/60 in the better eye. VI was defined as presenting VA of at least 3/60 but less than 6/12 in the better eye".
The authors table 4 aims to explore the impact of “Cataract, URE, Glaucoma, ARMD, Optic atrophy, Optic neuropathy, Diabetic retinopathy, Corneal scar, Others” on VI and blindness but as mentioned above (WHO guidelines) only the best one eye VA was reported for the analysis.
We know that most of these diseases can occur unilaterally/one eye one (Cataract, URE, Glaucoma, ARMD, Optic atrophy, Optic neuropathy, Diabetic retinopathy, Corneal scar) therefore the eye without the disease/pathology (best eye) was examined to observe the impact of the disease on the worse eye.Please explore in the discussion section.
Lines 248: Conclusion: This is misleading: “Cataract, glaucoma and URE were the major cause of VI and blindness in Lagos State.” Lagos state of Bonny Cantonment Lagos, Nigeria
Lines 249: Conclusion: “VI was more prevalent among males and those 60 years and above.” Does this statement refer to Table 2. If so, then the P value is non-significant at 0.122
Lines 249-250: Conclusion: “Early screening for 249 detection and management of cataract, URE and glaucoma is highly advised to reduce the burdens of VI.” Does the author mean treatment instead of screening as cataract can be only treated by surgery, URE by glasses (mostly) and glaucoma (medication/surgery).
Author Response
Reviewer 2 comments
We are grateful for the helpful feedback by the reviewers that helped us improve the quality of the manuscript. We carefully responded to all points and have modified the manuscript accordingly.
Reviewer’s comment
“Visual Impairment and Blindness among Patients at Nigeria Army Eye Centre, Bonny Cantonment Lagos, Nigeria” is an interesting and must needed study and I would like to commend the authors for their work.”
Response
We appreciate the acknowledgement of the work put into the study.
Reviewer’s comment
“Lines 53 onwards: While the authors in the introduction section mentioned the importance of economics/occupation/education on eye health, the current manuscript has not mentioned or explored the impact economics/occupation/education on their sample size, analysis or discussion”.
Response
We agree; however, there was no information on the education or occupation of the participants in their files or hospital record hence we could not relate that to their VI.
Reviewer’s comment
“Table 4 mentions the causes of VI/blindness as: “Cataract, URE, Glaucoma, ARMD, Optic atrophy, Optic neuropathy, Diabetic retinopathy, Corneal scar, Others”. Majority of these disease/pathology have no relation to VI/blindness in children, as supported by the findings in Table 5. Please explore childhood related VI/blindness factors if the children (8.8%) are included in this report, i.e., amblyopia, strabismus, ocular trauma, etc”.
Response
The data extracted for children had uncorrected refractive error (75%) as the major cause of VI/blindness as shown in Table 5. There was no subgroup (amblyopia, strabismus etc) for the causes of uncorrected refractive error in patients’ files.
Reviewer’s comment
“Please elaborate in the methods and discussion section how participants were segregated with co-morbidities, as diseases such as cataract and diabetic retinopathy can occur simultaneously occur in the same person.”
Response
The selection of primary cause of VI/blindness among patients with co-morbidities has been addressed under the data collection section. Line 102-103
Reviewer’s comment
“Lines 80-83: “This population was used to determine the sample size using a single population proportion formula. VI was considered as a 50% proportion of the population; the sample size was calculated using 95% confidence interval, 3% margin error, and 10% non-respond rates. Hence, the sample size was 664.”
Does this mean the study was underpowered as lines 122 stated that the study assessed 500 participants, if so, please mention in the limitation of the study”.
Response
The low number of participants against the calculated sample size has been addressed under the limitation section. Line 247-248
Reviewer’s comment
Lines 115: “Operational definition: Participants in the study had their vision status determined using the WHO categories of VI. "Blindness was defined as a presenting VA of less than 3/60 in the better eye. VI was defined as presenting VA of at least 3/60 but less than 6/12 in the better eye".
Response
The subgroups under the VI have been addressed in Table 2, line 138.
Reviewer’s comment
The authors table 4 aims to explore the impact of “Cataract, URE, Glaucoma, ARMD, Optic atrophy, Optic neuropathy, Diabetic retinopathy, Corneal scar, Others” on VI and blindness but as mentioned above (WHO guidelines) only the best one eye VA was reported for the analysis.
We know that most of these diseases can occur unilaterally/one eye one (Cataract, URE, Glaucoma, ARMD, Optic atrophy, Optic neuropathy, Diabetic retinopathy, Corneal scar) therefore the eye without the disease/pathology (best eye) was examined to observe the impact of the disease on the worse eye. Please explore in the discussion section.
Response
We considered only the better eye to classify participants into the various VI categories. Almost all our participants had conditions occurring binocularly hence we were unable to do such an analysis. We however appreciate the suggestion.
Reviewer’s comment
Lines 248: Conclusion: This is misleading: “Cataract, glaucoma and URE were the major cause of VI and blindness in Lagos State.” Lagos state of Bonny Cantonment Lagos, Nigeria
Response
We have edited the manuscript as suggested; Cataract, glaucoma and URE were the major cause of VI and blindness in Bonny Cantonment- Lagos State, Nigeria, line 250.
Reviewer’s comment
Lines 249: Conclusion: “VI was more prevalent among males and those 60 years and above.” Does this statement refer to Table 2. If so, then the P value is non-significant at 0.122
Response
This statement has been corrected. “VI was more prevalent in males than females; however, there was no significant difference between the two proportions. The prevalence of VI among age groups was more significant for those 60 years and above. Line 251-252.”
Reviewer’s comment
Lines 249-250: Conclusion: “Early screening for 249 detection and management of cataract, URE and glaucoma is highly advised to reduce the burdens of VI.” Does the author mean treatment instead of screening as cataract can be only treated by surgery, URE by glasses (mostly) and glaucoma (medication/surgery).
Response
This has been rephrased: line 253. “We conclude that frequent eye screening, for early detection, treatment and management of cataract, URE and glaucoma is highly advised to reduce the burden of VI. Also, a population-based study on the distribution and causes of VI in Lagos State is recommended.”
“Conclusion
Cataract, glaucoma and URE were the major cause of VI and blindness in Bonny Cantonment- Lagos State, Nigeria. VI was nearly equally prevalent among males and females. However, the prevalence among age groups was more significant for the different age groups. Regular eye screening, for early detection, treatment and management of cataract, URE and glaucoma is highly advised to reduce the burden of VI. Also, a population-based study on the distribution and causes of VI in Lagos State is recommended. Line 249 - 255
Reviewer 3 Report
This study is one institution-based cross-sectional research on the distribution and causes of visual impairment and blindness in South-west of Nigeria. There’re some limitations in this manuscript.
Major problem:
As described in the manuscript, Lagos is the most populous city in Africa, with a population of over 21 million. And the study population were patients attending the Nigeria Army Eye Centre in Lagos State, Nigeria, between January 2016 and December 2016 – an estimated total population of about 5500 patients. It seems that the Nigeria Army Eye Centre is not a big eye center in Lagos, and the patient data of this hospital can’t reflect the patient distribution in Lagos. The value of this hospital’s data is limited.
Minor problems:
1.How to define mild, moderate and severe visual impairment? I didn’t find the definition of those three grades of visual impairment.
2.In abstract, when URE was written at the first time, it should be presented completely rather than an abbreviation.
3. In method part, the sample size was estimated as 664. However, why the number of enrolled patients was 500 in the study?
Author Response
Reviewer 3 comments
We are grateful for the helpful feedback by the reviewers that helped us improve the quality of the manuscript. We carefully responded to all points and have modified the manuscript accordingly.
Reviewer’s comment
This study is one institution-based cross-sectional research on the distribution and causes of visual impairment and blindness in South-west of Nigeria. There’re some limitations in this manuscript.”
Response
We agree and have accordingly addressed them in the manuscript.
Reviewer’s comment
Major problem:
As described in the manuscript, Lagos is the most populous city in Africa, with a population of over 21 million. And the study population were patients attending the Nigeria Army Eye Centre in Lagos State, Nigeria, between January 2016 and December 2016 – an estimated total population of about 5500 patients. It seems that the Nigeria Army Eye Centre is not a big eye center in Lagos, and the patient data of this hospital can’t reflect the patient distribution in Lagos. The value of this hospital’s data is limited.
Response
We agree; hence we have included this as a limitation in our study so that readers can understand the context of the study. Lagos State is big and one could have expected more than the estimated patients population but low utilization of eye care services in Lagos State could be the reason for the low number recorded. We acknowledge that our study has some inherent limitations. Our data was collected from a single center, limiting our ability to generalize our findings to the entire population”. Also, conclusions made are limited to Bonny Cantonment only which is the biggest military cantonment in Nigeria.” Line 250
Reviewer’s comment
“Minor problems:
- How to define mild, moderate and severe visual impairment? I didn’t find the definition of those three grades of visual impairment.”
Response
This has been addressed. Table 2, line 138
- In abstract, when URE was written at the first time, it should be presented completely rather than an abbreviation.
Response
This has been addressed. Line 29
Reviewer’s comment
In method part, the sample size was estimated as 664. However, why the number of enrolled patients was 500 in the study?
Response
This limitation has been notified under the limitation section. Line 247
Round 2
Reviewer 1 Report
The authors have adressed to all comments. I have no further issues.
Author Response
There was no issue to resolve
Reviewer 2 Report
For table 3 please follow the same presentation as table 2.
Mild (less than 6/12 - 6/18)
Moderate (less than 6/18 - 6/60)
Severe (less than 6/60 – 3/60)
Blindness (less than 3/60 – NPL)
Author Response
Response to reviewers’ comments
Reviewer’s comment
For table 3 please follow the same presentation as table 2.
Mild (less than 6/12 - 6/18)
Moderate (less than 6/18 - 6/60)
Severe (less than 6/60 – 3/60)
Blindness (less than 3/60 – NPL)
Response
Table 3 has been modified to follow the same presentation as in Table 2 as suggested (see table 3).

Reviewer 3 Report
Authors have corrected some limitations, however, there are some problems in this manuscript.
Major problem:
1. This eye center is small, the value of the patient data is limited. We wish authors can provide some information of this eye center, for example, where are the patients from? How many people they serve? Hence, we can understand the background of these data.
2. In method part, authors described they enrolled patients with complete data. However, why all patients enrolled are visual impairment? No patients with normal visual acuity in this eye center. It’s not logic. I doubt the authenticity of this data.
Minor problem:
In abstract, “Conclusion: Cataracts, glaucoma and uncorrected refractive error (URE) were the major causes of VI and blindness in Lagos State.” The data of your hospital can’t reflect Lagos State directly. Please correct it.
Author Response
Response to reviewers’ comments Round 2
Reviewers’ comments
Major problem:
This eye centre is small, the value of the patient data is limited. We wish authors can provide some information of this eye centre, for example, where are the patients from? How many people they serve? Hence, we can understand the background of these data.
Response
The information about the eye centre has been modified in the method as written below:
The study was conducted in the Army Eye Centre located in the Bonny camp a military cantonment in Ikoyi, Victoria Island, Lagos state with majority of the residents in the military. The centre is a sub-unit of the Nigeria Military Hospital Lagos State. It has clinics for paediatrics, anterior segment, glaucoma, and optometry, among other subspecialties, to provide for the residents of the cantonment and its surroundings with tertiary eye care services (line 75 to 80). Majority of the patients that visit the centre are in the Nigeria Military service.
The small number (5500) of the total patients that visited in a year could be because the eye centre is a subunit of the military hospital and carter more for the military people than the public based on its location in the military camp.
Reviewers’ comments
In method part, authors described they enrolled patients with complete data. However, why all patients enrolled are visual impairment? No patients with normal visual acuity in this eye center. It’s not logic. I doubt the authenticity of this data.
Response
There were no patients with normal vision because the study was focused on only patients with presenting visual acuity of less than 6/12 at distance in the better eye. The information has been modified in the method (line 92 to 95).
Minor problem:
Reviewer’s comment
In abstract, “Conclusion: Cataracts, glaucoma and uncorrected refractive error (URE) were the major causes of VI and blindness in Lagos State.” The data of your hospital can’t reflect Lagos State directly. Please correct it.
Response
The statement has been modified as suggested (Line 255 to 256).
